# From Experiments to Discovery: A Principled Approach to Measuring How Well LLMs Do Science

**Kanishk Gandhi**\* **Michael Y. Li** \* **Lyle Goodyear** **Agam Bhatia**

**Louise Li** **Aditi Bhaskar** **Mohammed Zaman** **Noah D. Goodman**

Stanford University

## Abstract

Understanding the world and explaining it with scientific theories is a central aspiration of artificial intelligence research. Proposing theories, designing experiments to test them, and then revising them based on data are key to scientific discovery. Despite the promise of LLM-based scientific agents, no benchmarks systematically test their ability to propose scientific models, collect experimental data, and revise them in light of new data. We introduce `BoxingGym`, a benchmark with 10 environments for evaluating experimental design (*e.g.,* collecting data to test a scientific theory) and model discovery (*e.g.,* proposing and revising scientific theories). To enable quantitative and principled evaluation, we implement each environment as a generative probabilistic model with which a scientific agent can run interactive experiments. These probabilistic models are drawn from various real-world scientific domains ranging from psychology to ecology. To evaluate a scientific agent's ability to collect informative experimental data, we compute the expected information gain (EIG), an information-theoretic quantity which measures how much an experiment reduces uncertainty about the parameters of a generative model. A good scientific theory is a concise and predictive explanation. To quantitatively evaluate model discovery, we ask a scientific agent to explain their model and evaluate whether this explanation helps another scientific agent make more accurate predictions. We evaluate several open and closed-source language models of varying sizes. We find that larger models (32B) consistently outperform smaller variants (7B), and that closed-source models generally achieve better results than open-source alternatives. However, all current approaches struggle with both experimental design and model discovery, highlighting these as promising directions for future research. [2]

> "To understand a system, you must perturb it."
> – George Box (*ad sensum*)

## 1 Introduction

Helping humans understand the world (and themselves) by discovering scientific theories is a foundational goal of artificial intelligence research [30]. Proposing theories about the world, conducting experiments to test them, and revising them based on data is central to this process [9]. Recent advances in large language models (LLMs), have shown promising potential for accelerating scientific discovery. LLMs have extensive scientific knowledge [2], strong inductive reasoning capabilities [52, 42], and the ability to propose models of data [26, 27, 11]. These promising results suggest that LLMs, functioning as autonomous agents, could be well-suited for experimental design (*i.e.,* collecting informative experiments to test scientific theories) and model discovery (*i.e.,* developing interpretable models based on experimental data).

---

\*Equal Contribution. Corresponding author: `kanishk.gandhi@stanford.edu`
[2]Project: https://github.com/kanishkg/boxing-gym/

Submitted to 39th Conference on Neural Information Processing Systems (NeurIPS 2025). Do not distribute.

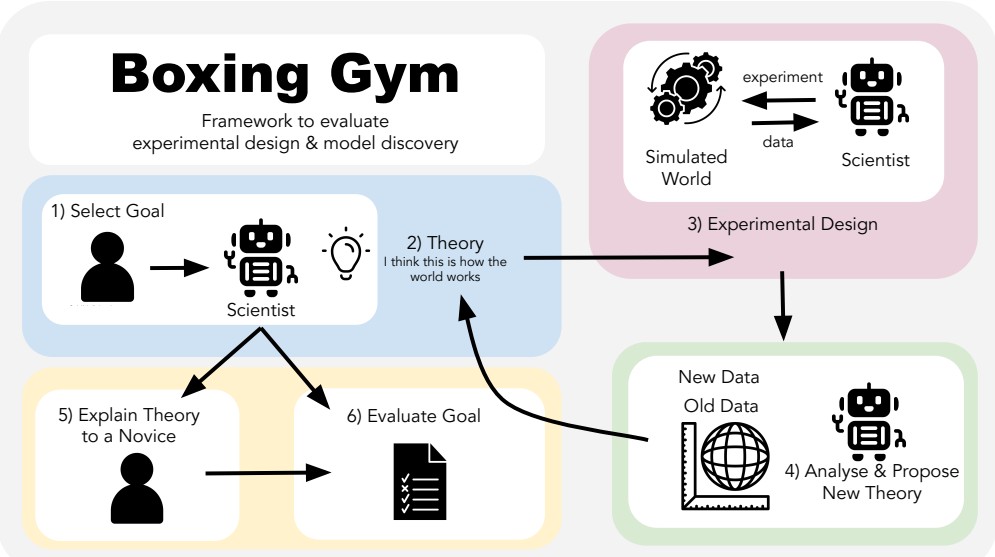

Figure 1: **Overview of** `BoxingGym`**.** The `BoxingGym` Framework is designed to holistically evaluate experimental design and model discovery capabilities in the spirit of George Box [9]. 1) The process starts with a user defining a goal for the scientist agent. 2) The scientist formulates a theory. 3) This theory guides the experimental design, where the scientist interacts with a simulated world to gather new data. 4) The scientist then analyzes the new and old data to propose and refine theories. This iterative process continues for several iterations. 5) The scientist is then asked to explain the findings to a novice. 6) We evaluate the novice and the scientist by casting the goal as a prediction problem.

Previous work has evaluated automated experimental design and model discovery in isolation [16, 17, 15, 26]. However, they are fundamentally coupled in real-world settings: scientists collect experimental data to build better models and better models inform better experiments. While scientific agents are promising, there is currently no systematic way to evaluate an agent's ability to propose scientific models, collect experimental data, and revise them in light of new data. This motivates the need for a benchmark that evaluates an agent's capabilities holistically in an integrated scientific discovery pipeline.

We outline the key desiderata for a framework that evaluates experimental design and model discovery: (1) The framework should enable the agent to *actively experiment* with the environment without requiring the agent to perform time-consuming and resource-intensive real-world lab experiments. (2) Since scientific theories come in different forms, the framework should flexibly accommodate *different representations of scientific theories*. (3) The framework should evaluate experimental design and model discovery in an *integrated* way. (4) Science is often *goal-directed* or driven by an inquiry. For example, a biologist might perform experiments with the goal of identifying cellular mechanisms underlying circadian rhythm in mammals. Our framework should allow users to specify high-level goals to guide the agent's discovery process. Our desiderata are inspired by the framework for scientific modeling introduced by George Box [7, 8], which emphasizes an iterative process of building models, designing experiments to test them, and revising them accordingly.

To achieve these desiderata, we introduce `BoxingGym` (Fig. 1) a flexible framework for evaluating experimental design and model discovery with autonomous agents. Our benchmark consists of 10 *environments* grounded in real-world scientific models. To enable agents to actively experiment, we implement each environment as a generative model. This key design choice makes simulating active experimentation tractable because it corresponds to sampling from the underlying generative model, conditioned on the experimental interventions. To accommodate various representations of scientific theories, all environments are designed with a flexible language based interface (Fig. 2). Finally, our environments can be instantiated with different goals, or intents for inquiry, that encourage the agent to adapt their experimentation towards accomplishing the goal (*e.g.,* understand the parameters underlying participant behavior in a psychology study) by specifying the goal in language.

We introduce principled evaluation metrics that measure the quality of experiments and discovered models. To evaluate experimental design, we draw from *Bayesian optimal experimental* (BOED)

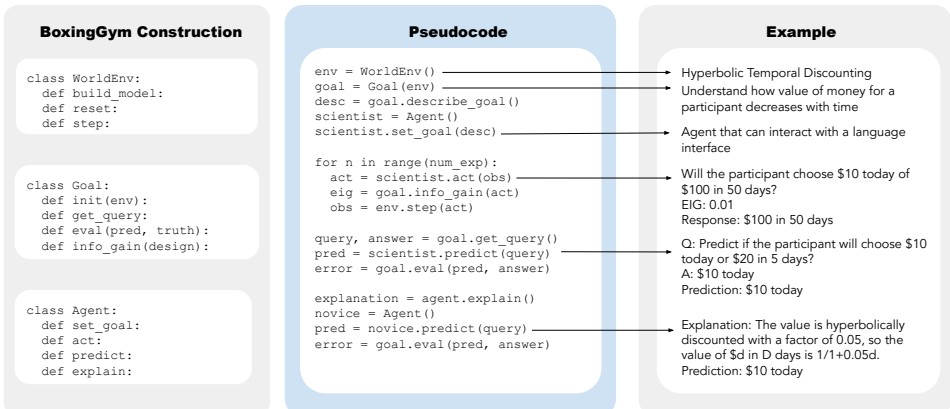

Figure 2: **Python pseudocode examples. (left)** `BoxingGym` is instantiated as modular classes and methods for the environment (WorldEnv), goals (Goal), and agents (Agent). **(center)** Pseudocode illustrating the workflow of setting goals, performing experiments, predicting outcomes, and providing explanations. **(right)** An example, hyperbolic temporal discounting, where the agent predicts a participant's choice between immediate and delayed rewards and explains the concept to a novice.

design [43] and use *expected information gain* (EIG) to measure the informativeness of an experiment. EIG captures how much an experiment reduces uncertainty in the parameters of a generative model and, importantly, this measure complements our decision to implement environments as generative models. To evaluate model discovery, we take inspiration from the fact that science is a communicative endeavor. We propose a *communication-based* evaluation strategy: we ask a scientist agent to distill their experiments into a natural language explanation and evaluate how much that explanation empowers a novice agent, who does not have access to the experiments conducted by the scientist, to make accurate predictions about the environment.

We evaluate several open and closed-source language models ranging from 7B to 32B parameters. We find that larger models consistently outperform smaller variants, and closed-source models generally achieve better results than open-source alternatives. We also evaluate Box's Apprentice [26], which augments language models with statistical modeling capabilities, but find that this augmentation does not reliably improve performance. Notably, we observe substantial variation in difficulty across environments, which remaining challenging even for the strongest models. Promisingly, some environments show clear performance improvements with model scale. These results highlight significant opportunities for improving automated scientific reasoning.

## 2 Related Works

**Optimal Experimental Design.** Bayesian optimal experimental design (BOED) is a principled framework for designing maximally informative experiments across various disciplines [48, 12, 34]. While theoretically appealing, BOED's practical implementation is challenging due to the intractability of information gain metrics like expected information gain (EIG). Although several methods [43, 16, 17] exist to approximate EIG, they assume the data follows a fixed generative model—limiting their utility when model revision is needed as new data is collected.

**Automated Model Discovery.** Automated model discovery from data has been a long-standing goal in AI, aiming to build interpretable models that capture underlying patterns in data—from physical laws [6, 31] to nonparametric regression [15]. Recent work [26, 27] has integrated language models into this process, leveraging their ability to both propose and critique candidate models, demonstrating their potential as tools for automated model discovery. This work highlights the potential of using language models as a powerful tool for model discovery.

**Reasoning and Exploration with LLMs.** Language models have shown promising capabilities in both deductive reasoning (deriving consequences from hypotheses) Saparov et al. [46], Saparov and He [45], Poesia et al. [41] and inductive reasoning (inferring hypotheses from observations) [52, 42]. While reinforcement learning has improved LLMs' reasoning abilities [23, 21, 20, 22], these advances have primarily focused on deterministic, verfiable systems rather than the stochastic data typical in scientific discovery. Efficient exploration and information-seeking are crucial for experimental design and model building. Recent work [36, 32, 19, 18, 47, 25] has investigated in-context exploration

102 strategies and shown how language models can learn how to search and explore directly through
103 sequence modeling, developing effective search strategies in language.

104 **Interactive Environments.** Drawing inspiration from established reinforcement learning principles
105 [10, 33], BoxingGym adopts the modularity and simplicity of classic environments like OpenAI Gym
106 while shifting focus to evaluation rather than agent training. While recent work has expanded interac-
107 tive benchmarks to language agents —spanning tasks from software debugging [24] to automated
108 scientific research[35, 28], our work advances this direction by introducing a principled framework
109 for evaluating language agents' capabilities in iterative experimental design and model discovery.

# 3 Boxing Gym

## 3.1 Problem Formulation.

112 We formalize experimental design and model discovery using probabilistic modeling and Bayesian
113 optimal experimental design (BOED). In BoxingGym , each environment is implemented as a
114 generative model defining a joint distribution over the experimental outcome $y$, experimental design
115 $d$, and unobserved parameters $\theta$. This joint distribution is defined in terms of a prior distribution
116 over $\theta$, $p(\theta)$ and a *simulator* $p(y|\theta, d)$ which is a model of the experimental outcome $y$ given
117 parameters $\theta$ and design $d$. For example, in a psychology experiment, $\theta$ could be the parameters
118 of a behavioral model of participants, $d$ could be the questions posed to participants, and $y$ could
119 be the participant's response to $d$. Running an experiment corresponds to choosing a design $d$
120 and observing a sample $y$ from the marginal predictive distribution conditioned on that design,
121 *i.e.,* $y \sim p(y|d) = E_{p(\theta)}[p(y|\theta, d)])$ [3].

## 3.2 Evaluation

### 3.2.1 Evaluating experimental design via Expected Information Gain

124 To evaluate experimental design, we take inspiration from the Bayesian OED literature [16, 17].
125 Crucially, our choice to implement environments as generative models enables us to leverage this
126 literature. For each domain, we have an underlying predictive model $p(y|\theta, d)$. We quantify the
127 *informativeness* of a design $d$ through the expected information gain (EIG), that measures the
128 reduction in posterior uncertainty about the model parameters $\theta$ after running an experiment $d$. Below,
129 $H$ is the Shannon entropy.

$$\text{EIG}(d) = \mathbb{E}_{p(y|d)}\left[H[p(\theta)] - H[p(\theta|y, d)]\right]$$

130 Since the EIG is typically not available in closed-form, we use a Nested Monte Carlo estimator

131 $$\hat{\mu}_{\text{NMC}}(d) = \frac{1}{N}\sum_{n=1}^{N}\log\left(\frac{p(y_n|\theta_{n,0}, d)}{\frac{1}{M}\sum_{m=1}^{M}p(y_n|\theta_{n,m}, d)}\right) \quad \text{where} \quad \theta_{n,m} \overset{\text{i.i.d.}}{\sim} p(\theta),\ y_n \sim p(y|\theta = \theta_{n,0}, d)$$

132 We chose this estimator because it is a consistent estimator of the true EIG [43] and is straightforward
133 to implement. EIG measures the value of an experiment under the assumption that the true distribution
134 of experimental outcomes is modeled by $p(y|d)$. In general, this assumption is not true, but EIG is
135 still a useful measure since we generate data from an underlying model in our benchmarks.

### 3.2.2 Evaluating model discovery via communication

137 To evaluate the quality of a model, we use standard model evaluation metrics (*e.g.,* prediction MSE)
138 and a communication-based metric that takes advantage of the natural language interface. In particular,
139 a *scientist agent* interacts with an environment through experiments. After these experiments, we ask
140 the scientist agent to synthesize their findings through an *explanation*. We then evaluate how much
141 that explanation enables a *novice* agent to make more accurate predictions about the environment
142 without any additional experiments. Since a good explanation is both *predictive* and *parsimonious*,
143 we set a token limit on the explanation. Crucially, this evaluation method can accommodate different
144 forms of scientific theories. In our experiments, we ask the scientist agent to produce a statistical
145 model and then distill the model into a natural language explanation to guide the novice agent.

---

[3]In the sequential setting, we replace the prior $p(\theta)$ with the posterior $p(\theta|y, d)$.

### 3.2.3 Evaluating goals via prediction

To evaluate success at achieving a specific goal (*e.g.,* how do the populations of predator and prey change with time) we employ a prediction target (*e.g.,* predict the population of predators at a particular time) and calculate a standardized prediction error. First, we compute the error between the predicted and true values. Then, we standardize this error with respect to the prior predictive mean, which is obtained by assuming a uniform prior over the design space. Specifically, for each domain, we sample a design $d$ uniformly from the design space and a parameter $\theta$ from the prior distribution $p(\theta)$. We then generate samples from the predictive model $p(y|\theta, d)$ and average over multiple $d$ and $\theta$ to obtain the prior predictive mean $\mu_0$ and variance $\sigma_0$. Let $\{y_i\}_{i=1}^n$ be the ground truth outputs for inputs $\{x_i\}_{i=1}^n$. and let $\{\hat{y_i}\}_{i=1}^n$ be the predictions of the agent. The standardized prediction error is then calculated using these quantities, providing a measure of the agent's performance relative to the prior predictive mean. We use a domain-specific function $f$ computing the discrepancy between a prediction $\hat{y_i}$ and ground truth value $y_i$ (*e.g.,* MSE). We compute the errors $\epsilon_i = f(\hat{y_i}, y_i)$ and $\epsilon_{\mu_0} = f(\mu_0, y_i)$. Finally, we compute the standardized error as $\frac{\epsilon_i - \epsilon_{\mu_0}}{\sigma_0}$. Crucially, since this metric is computed with respect to the prior predictive, this metric can be negative.

### 3.3 Design Decisions in Constructing `BoxingGym`

We outline the key design decisions of `BoxingGym` that allow it to capture key aspects of scientific discovery within a flexible, simulated, and extensible environment.

**Discovery via active experimentation.** The agent actively interacts with the environment by conducting experiments, reflecting the real-world coupling of experimentation and model discovery. This approach assesses the agent's ability to gather relevant data and refine its models based on experimental results.

**Real-world scientific models.** Our environments are grounded in real-world scientific models from several domains, ensuring the benchmark tests the agent's ability to handle realistic scenarios. We implement these environment as `pymc` generative models to make active experimentation an automatic and tractable process.

**Goal-driven discovery.** Each environment has a specific goal, mirroring the inquiry-driven nature of scientific research. This encourages the agent to engage in targeted experimentation.

**Language-based interface for experiments.** We use a language-based interface for our experiments because it's flexible (*i.e.,* scientific domains can generally be described in language), easily integrates with LLMs, and interpretable to humans.

**Emphasis on Measuring Discovery with Explanations.** `BoxingGym` places a strong emphasis on measuring the quality of the agent's discoveries through the explanations it can provide after experimentation (§3.2.2). This design decision is motivated by two considerations. From a theoretical perspective, science is fundamentally about developing better theories, and scientific theories are explanations of observed phenomena. From a practical perspective, communicating findings to the broader scientific community is an essential aspect of scientific research. By using language, we do not have to commit to a particular representation of a scientific theory. We illustrate this flexibility, by showing how different representations can be easily integrated within our method for measuring natural language explanations.

**Extensible/modular environments for benchmarking agents.** `BoxingGym` is easily extensible and modular, enabling researchers to integrate new environments and test different agents with minimal effort. We illustrate this in Fig. 2 which provides a pseudo-code example of how to implement a new environment and goal in `BoxingGym` .

### 3.4 Domains

`BoxingGym` consists of 10 environments (see App. D for full details) that cover a range of scientific domains and test different aspects of experimental design and model discovery. Some environments are designed to test optimal experiment design, while others focus on model discovery or involve simulated neuro-symbolic human participants.

**Location finding.** [17] In an $n$-dimensional space with $k$ signal-emitting sources, the scientist measure signals at any grid location. Goals include predicting the signal at any point or locating the sources.

**Hyperbolic temporal discounting.** [17] The scientist observes a participant's choices for different immediate rewards ($ir$), delayed rewards ($dr$), and delay periods ($D$ days) Fig. 2 (right). Goals include predicting choices of a participant or discount factors.

**Death process.** [17] A disease spreads at an infection rate. The scientist can measure the number of infected individuals at different points of time to predict future infections or the infection rate.

**Item Response Theory (IRT).** [44] In this environment, there is a set of students and a set of questions. The experimenter can observe the correctness of a student's response to a particular question. The goal is to discover the underlying model that relates student ability and question difficulty to the probability of a correct response.

**Animal growth curves.** [29] An experimenter can observe the length of a dugong at a particular age. The goal is to discover the underlying growth model of dugongs.

**Population growth dynamics.** [29] An experimenter can observe the population of peregrines at a particular point in time. The goal is to discover the underlying population dynamics model. This is tested by asking the experimenter to predict population dynamics at a particular point in time.

**Mastectomy Survival analysis.** [13] The experimenter can observe if a patient is alive after a mastectomy, including metastasis status and time since surgery. The goal is to predict survival probabilities for new patients.

**Predator-Prey dynamics.** [51] This simulates predator-prey populations over time. The goal is to discover models like the Lotka-Volterra equations to predict future populations.

**Emotion from outcome.** [37] Participants guess a player's emotions after a gambling game's outcome. The experimenter designs games with varied probabilities and prizes to model how participants judge the emotions of a player from outcomes. Human participants are simulated using a probabilistic model translated into natural language by a language model.

**Moral Machines.** [5] Participants face moral dilemmas, choosing which group an autonomous car should save. Experimenters manipulate group compositions and required actions to model moral decision-making. Human participants are simulated with a probabilistic model, and their actions are translated into natural language by a language model.

## 4 Experiments

We conduct experiments to evaluate the performance of two baseline agents on BoxingGym . Our goal is to assess their ability to perform experimental design and theory building across a diverse set of environments. We benchmark two types of agents: a standard language model (GPT-4o, OpenAI [38]) and a language model augmented with symbolic reasoning capabilities (Box's Apprentice).

**LLM Agent.** We consider 6 LLMs, GPT-4o [38], Claude-3.7-sonnet [3], Qwen-2.5-32b-instruct, Qwen-2.5-7b-instruct [54], and reasoning variants OpenThinker-32b, and OpenThinker-7b [50]; the reasoning variants are finetuned on math and coding task. We prompt these models to interact with our environment, purely through natural language, without additional tools (see Fig. 2, see App. B for details).

**Box's Apprentice.** The apprentice agent augments language models by enabling them to implement generative models of observed data. For model discovery, the agent writes a pymc program [26] after 10 experiments, which is then fit and provided to the scientist explaining findings to the novice. For experimental design, the agent creates and uses these models to guide subsequent experiments.

**Experiment Setup.** For each environment, we run the agents for 5 independent trials. At each step, the agent chooses to perform an experiment, by specifying a design, and observes the outcome.

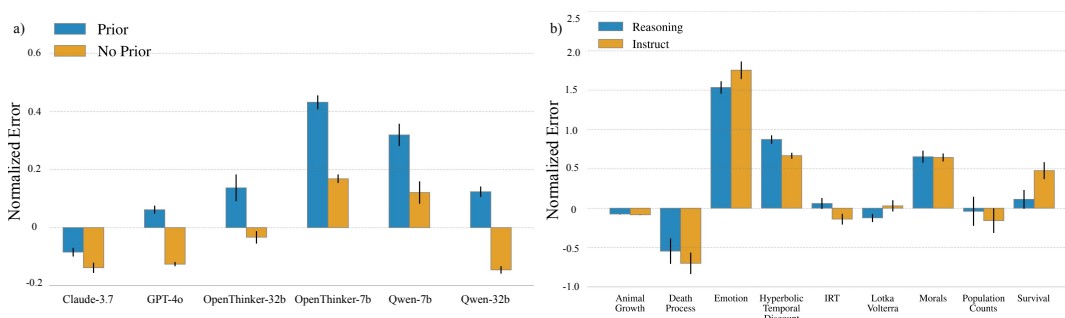

Figure 3: **Normalized Error Compared across Models.** (a) Comparison of the normalized errors for different LLMs with or without prior information included in the prompt. (b) Comparison of reasoning models (OpenThinker) and instruct models (Qwen) across environments. Error bars are the standard error across 5 runs.

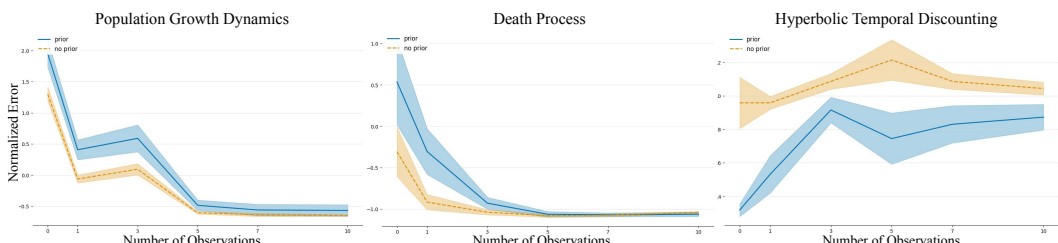

Figure 4: **Normalized Errors Over Number of Observations.** Normalized errors for the LLM agent with `gpt-4o` with prior information (solid blue) and without prior information (dotted yellow) across three domains: Population Growth Dynamics (left), IRT (center) and Hyperbolic Discounting (right). Error bars are the standard error across 5 runs.

After a fixed number of steps (0, 1, 3, 5, 7, 10), we evaluate the agent's performance using the metrics described earlier §3.2. The performance of models is averaged across 5 runs and over 10 evaluation points. We also explore a *prior* vs *no prior* condition to investigate whether domain knowledge helps or hinders scientific discovery. In the prior condition, we give the LM full context about the problem domain (*e.g.,* "you are observing how participants balance delayed vs immediate rewards"), simulating scientists with background knowledge. In the no prior condition, we remove this context and describe the setting in a domain-agnostic way (*e.g.,* "you receive a tuple of three values"), resembling reasoning from raw observations without preconceptions. This tests whether prior knowledge scaffolds discovery or creates biases that constrain exploration.

## 4.1 Experimental Design Evaluation

**Setup.** To evaluate the agents' performance, we first assess their ability to gather valuable information through their experiment selection and then measure how effectively they use this information to predict the environment. The Expected Information Regret (EI Regret) compares the Expected Information Gain (EIG) (§3.2.1) of the agent's chosen experiments to the maximum EIG achievable from 100 random experiments. Lower EI Regret indicates more informative experiment selection.

**Prior information does not improve performance.** We find that models often perform better when given no prior information after 10 experiments (Fig. 3a). In some cases, this is because the LLM makes an overly strong assumption about the environment (*e.g.,* the signal decay is symmetric around the origin) and does not revise the assumption after more experiments; this is consistent with findings reported by Li et al. [26]. In other cases, such as the hyperbolic discounting environment (Fig. 4, right), the model overfits to limited observations.

**More experiments generally lead to better predictions.** We plot the learning trajectories for three environments in (Fig. 4). The agent's average prediction error decreases as it performs more

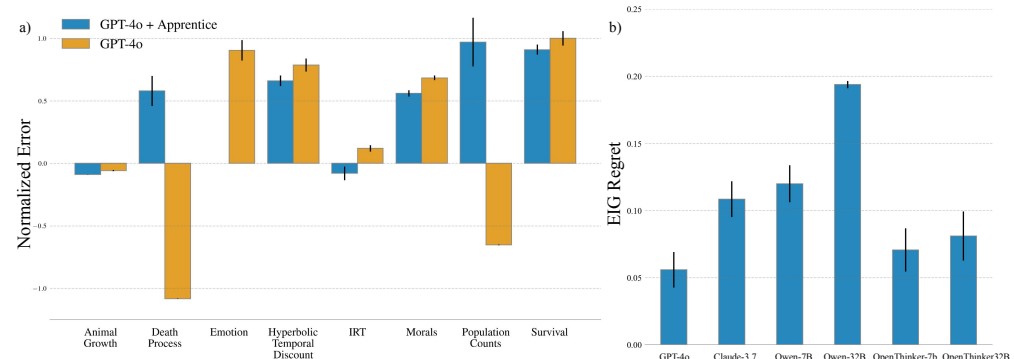

Figure 5: (a) Comparison of the Box's Apprentice with an LLM agent. (b) EIG Regret scores for six large language models, with lower values indicating better performance.

experiments. The Hyperbolic Temporal Discounting environments shows an unexpected trends where more experiments actually increases error. This may again be related to how prior knowledge interferes with effective learning from data.

**Models Improve with Scale.** Larger models consistently outperform their smaller counterparts within the same model family. Both OpenThinker-32B and Qwen2.5-32B demonstrate significantly better performance than their respective 7B variants across environments (Fig. 3a), highlighting the benefits of scale for experimental design tasks.

**Instruction-Tuned Models outperform Reasoning Models.** Surprisingly, the instruction-tuned Qwen2.5 models outperform the reasoning-focused OpenThinker models (Fig. 3b). This may be because OpenThinker models are finetuned to perform well on a relatively narrow set of verifiable problems in math and code, while instruction-tuned models retain broader capabilities that could be useful for experimental design.

**Models performance varies substantially across environments.** Models show varying performance across different environments (Fig. 3b). Performance is strongest on environments like population growth dynamics and death process, where the LM agent achieves negative standardized error, indicating that the LM successfully leveraged information gained through experimentation. However, in environments like hyperbolic discounting, performance is low even after experimentation, suggesting that some domains are inherently more challenging for current models.

**EIG Regret reveals relationship between experimental design and prediction.** Our EIG regret analysis (Fig. 5b) provides insight into the relationship between two key components of scientific reasoning: designing informative experiments and making accurate predictions from collected data. GPT-4o achieves both the lowest EIG regret and strong predictive performance across several environments, suggesting these capabilities can be aligned. However, the varying performance of other models is informative — for instance, Qwen-32B shows higher EIG regret despite good predictive performance in some domains, indicating that while these abilities may be related, excellence in prediction doesn't automatically translate to optimal experimental design.

**LLMs cannot always optimally leverage statistical models.** While Box's Apprentice can propose and fit explicit statistical models to observed data, it does not consistently improve over the non-augmented LLM (GPT-4o) (Fig. 5a) From qualitative analysis of the models, we find that Box's Apprentice tends to favor overly simple functional forms due to limited data, such as using linear approximations for inherently nonlinear phenomena.

### 4.2 Evaluating Model Discovery via Communication

**Setup.** Next, we evaluate the agents' ability to build and communicate models that capture the underlying phenomena in each environment. To test this, we have the agents interact with the environment for 10 steps (scientist phase) and then generate a natural language explanation of their

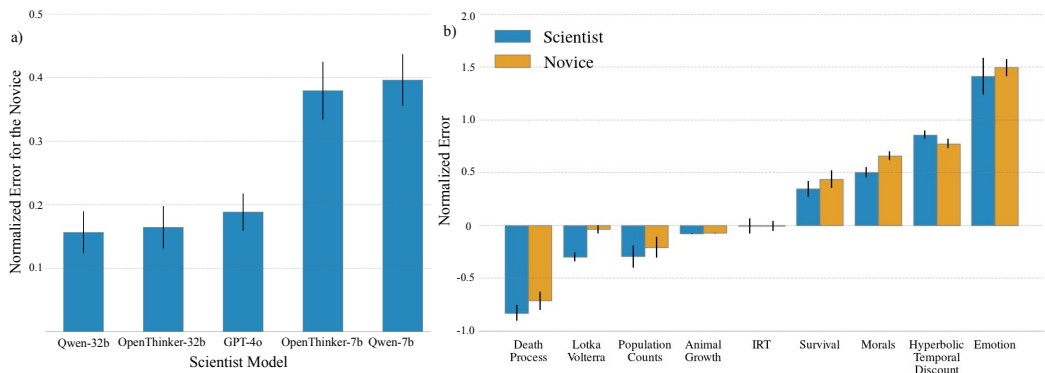

Figure 6: **Evaluation of Model Discovery via Communication.** (a) Comparison of the standardized error of the Novice (`gpt-4o`) with different Scientist models. (b) Comparison of errors made by the Novice and the Scientist (both models are `gpt-4o`). Error bars are standard error.

findings. We then provide this explanation to a *novice* agent, which must make predictions about the environment without any direct interaction (novice phase by using the explanation from the scientist; §3.2.2). The novice agent is always `gpt-4o`. The scientist's prediction after 10 observations (Error After Experiments) acts as a weak positive control. Ideally, if the scientist's explanation is effective, the novice's error should approach the positive control.

**Explanations improve with scale.** Larger models generally produce more effective explanations, as evidenced by better novice performance when using explanations from 32B variants compared to 7B models (Fig. 6a). This suggests that increased model scale improves not just experimentation but also the ability to distill and communicate findings.

**Explanations are not as good as experiments** As expected, novice agents perform worse than scientists who directly interacted with the environment (Fig. 6b). The gap suggests that current explanation methods do not fully capture the knowledge gained through experimentation.

**Explanations are more helpful for some environments.** However, the effectiveness of explanations varies substantially across domains (Fig. 6b). For instance, explanations are helpful for animal growth, but struggle with complex domains like moral judgments. This variation likely reflects the complexity of different domains and the current limitations of language models in capturing and communicating certain types of patterns.

## 5 Discussion

We introduced BoxingGym , a benchmark measuring language-based agents' capabilities in experimental design and model discovery across 10 real-world-based environments. We evaluated experimental design using information gain metrics and developed a novel model discovery metric based on an agent's ability to explain its model to a novice agent. Our evaluation across multiple model scales (7B-32B parameters) shows that while larger and closed-source models generally perform better, fundamental challenges persist. Neither domain-specific prior knowledge nor statistical modeling capabilities consistently improved performance. Some environments yielded strong results with larger models, while others remained challenging for all approaches. BoxingGym has limitations: it uses pre-defined experimental paradigms rather than requiring design from scratch [14], ignores resource constraints, and covers limited scientific domains. Future work should address these limitations by incorporating experiment design from scratch, resource constraints, and more diverse fields. We could also expand the human behavior environments (Moral Machines, Emotions) with more sophisticated participant simulations [4, 1, 49, 39, 40]. While our experiments demonstrated potential for interfaces that augment language models' scientific reasoning capabilities, future research should explore data visualization, strategic simulations [27], model validation, and web-based research strategies to enhance experimental guidance and discovery.

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
