# OpenReview forum: "From Experiments to Discovery: A Principled Approach to Measuring How Well LLMs Do Science"
_NeurIPS.cc/2025/Datasets_and_Benchmarks_Track — Submitted to NeurIPS 2025 Datasets and Benchmarks Track_

### Official Review · Reviewer_eHZa · 2025-06-17

**Rating:** 2
**Confidence:** 3

**Summary:**

The paper introduces boxingGym, a benchmark that **supposedly** tests LLMs' performance at conducting scientific research. I cannot see obvious contributions from the paper.

**Dataset Code Accessibility:**

Yes

**Dataset Code Comments:**

Repo is provided. I did not test it myself.

**Ethical Considerations:**

No, there are no or only very minor ethics concerns

**Final Justification:**

The authors argued that an important contribution is a mechanism that "discovers" abstract concepts and symbolic equations. First, this contribution is not being discussed or evidenced in the paper itself. Second, in the one evidence through response, it is not clear if the AI invents abstract concepts (which is a key ingredient of doing science) or fetched an existing concept from its knowledge base.

**Limitations Weaknesses:**

1. From supplementary, all tasks are about building simple statistical models through active learning. It would be much more convincing if the paper can include more representative scientific modeling/discovery tasks where some of the followings are present: (1) multi-physics at multiple time and length scales are coupled (e.g., chemical engineering, life sciences, and material sciences), (2) scalability becomes a bottleneck (e.g., molecular dynamics and DFT), (3) insights are only drawn from solving inverse problems (e.g., optimal design, strategic planning, conjecture proposing and proof derivation).

2. LLMs are only used for generating analysis code and interpreting numerical results. The only performance metric used is MSE. The paper would be more attractive if LLMs are tested on multimodal science discovery capabilities such as (1) derivation of governing equations from first principles and data, (2) strategic model segmentation across time and length scales, and model reduction based on dominating disciplines, etc.

**Strengths Contributions:**

The idea of benchmarking LLMs on various scientific discovery tasks is timely, yet the execution of this paper is far from satisfactory.

---

> ### Author Rebuttal · Authors · 2025-07-31
>
> Thank you for your feedback. We address your concerns:
>
> > Tasks are simple statistical models … more representative scientific modeling/discovery tasks … multi-physics at multiple time and length scales are coupled (e.g., chemical engineering, life sciences, and material sciences)
>
> While our environments use statistical models, the challenges are far from simple:
> - Current SOTA fails: Even Claude-3.7-sonnet fails to improve over simple baselines,
> - Fundamental capabilities tested: These seemingly simple tasks require hypothesis formation, experimental design, and belief revision - core to all scientific discovery
> - Real-world grounding: Our models, albeit simple, come from actual scientific domains (ecology, cognitive science, epidemiology)
>
> We show that LLMs struggle even with these foundational tasks, making more complex environments premature. We agree that expanding the tasks to more complex, even multimodal ones is exciting future work. Our framework is flexible enough to simulate and implement more complex, higher dimensional models, and this is an active area of work to extend it. The benchmarks we use in our paper serve as a good starting point that already highlights key limitations of these models.
>
> We will clarify this in the discussion, highlighting limitations and potential future work.
>
> > Only one metric (MSE)
>
> We actually use three metrics:
> - MSE: For prediction error (generalizable across domains). We also standardize this measure so that prediction errors can be directly compared across environments.
> - EIG Regret: For experimental design quality
> - Communication score: While this uses predictive error with MSE, the overall evaluation framework requiring communication goes beyond measuring predictive error.
>
> Finally, domain specific metrics can easily be incorporated into our gym environments as needed.
>
> Please let us know if you have any other questions.

---

> > ### Comment · Reviewer_eHZa · 2025-08-01
> >
> > Author responses are tangential to my comments: My main point is that the case studies should expose a full spectrum of open challenges in attempts to automate scientific discovery so that follow-up studies can make meaningful progresses. Simply showing that publicly available AI models cannot do a subset of simple tasks is missing the point.
> >
> > I believe the following more concrete request would be reasonable based on what the title claims:
> >
> > Consider the task of developing a new material, within which new physics (not statistical) models and explainable knowledge (not model parameters) explaining process-structure-property relationships have to be derived. A human effort includes a dynamic process for identification of key modeling aspects (e.g., process steps and ingredients, defect types, butterfly effects within certain length and time scales) based on prior knowledge, analytical deduction, hypothesis generation and testing, etc.
> >
> > Please explain how BoxingGym measures, in a principled way, "how well LLMs perform this task" by analyzing the value of LLM decisions along a similar dynamic process.

---

> > > ### Author Response · Authors · 2025-08-04
> > >
> > > Thank you for the clarification. We'll address how BoxingGym captures the scientific discovery process meaningfully:
> > >
> > > **Statistical Models Require Understanding Underlying Mechanisms**
> > >
> > > Consider the Hyperbolic Temporal Discounting environment. While implemented as a generative model, discovering the correct theory requires the same scientific reasoning as any discovery process:
> > > - Discovering Latent Concepts: Agents must invent the abstract concept of a "discount factor" (k) to explain human choices; this latent variable isn't directly observable but provides crucial mechanistic insight. This mirrors how scientists must discover hidden variables (e.g., pressure explaining gas behavior, or cognitive load explaining memory performance).
> > > - Theory Formation and Testing: Success requires:
> > >     - Hypothesizing competing mechanisms (exponential vs. hyperbolic vs. quasi-hyperbolic discounting)
> > >     - Designing discriminative experiments that maximize information between theories (selecting reward/delay combinations that best distinguish models)
> > >     - Discovering the governing equation
> > > - Mechanistic Understanding: Agents must recognize that k represents individual time preference, a cognitive mechanism explaining phenomena like preference reversals and procrastination, not just curve-fitting observed choices.
> > >
> > > While our environments use statistical models for tractable simulation, discovering the correct model requires understanding the underlying mechanisms.
> > >
> > > We acknowledge that BoxingGym doesn't capture all aspects of scientific discovery, but it does evaluate the core cognitive capabilities: forming hypotheses, designing discriminative experiments, updating beliefs based on data, and synthesizing mechanistic explanations. We'll revise our paper to better emphasize this.
> > > Please let us know if this clarifies our contribution and if you have any other questions.

---

> > > > ### Comment · Reviewer_eHZa · 2025-08-05
> > > >
> > > > Please kindly help me better understand these arguments from your response:
> > > >
> > > > 1. "Agents must invent the abstract concept of a "discount factor" (k) to explain human choices", "Discovering the governing equation"
> > > > - Where in the paper is this invention of abstract concepts discussed with quantitative evidence? My understanding thus far is that the automated learning process produces models that predict, but are not necessarily similar to the GTs in their functional forms. If it is the symbolic equations being automatically discovered, that certainly changes the whole story. So please clarify.
> > > >
> > > > 2. "Hypothesizing competing mechanisms..."
> > > > - Again, please point to the section in the paper (or supplementary) where the model demonstrates the ability to form hypotheses by itself. Sorry if I missed it!
> > > >
> > > > 3. "Designing discriminative experiments that maximize information between theories"
> > > > - This is just statistics, nothing new.
> > > >
> > > > 4. "Agents must recognize that k represents individual time preference, a cognitive mechanism explaining phenomena like preference reversals and procrastination, not just curve-fitting observed choices."
> > > > - First, is the agent already aware of the existence of "k" and/or the governing equation? That would contradict the claim that the agent **invents** abstract concepts. Then, please point to the section in the paper where the agent recognizes the cognitive meaning of "k". Further, for the no-prior setting, how would the agent come up with a concept related to cognition?
> > > >
> > > >  5. "it does evaluate the core cognitive capabilities..."
> > > > - This is by itself a strong hypothesis. Does an established behavioral scientist have the core cognitive capabilities to contribute meaningfully as a mathematician or physicist? I doubt.

---

> > > > > ### Author Response · Authors · 2025-08-06
> > > > >
> > > > > Thank you for your prompt response and continuing the discussion.
> > > > >
> > > > > > First, is the agent already aware of the existence of "k" and/or the governing equation?
> > > > >
> > > > > **The agent starts with no knowledge of the discount factor concept or governing equations.** In the hyperbolic temporal discounting environment with priors, the agent only knows: *"A person has to choose between a delayed reward dR dollars in x days and an immediate reward iR dollars today."* The agent is not told that:
> > > > > - A discount factor exists or would be useful
> > > > > - How temporal preferences might be modeled mathematically
> > > > > - What functional form governs the choices
> > > > >
> > > > > Instead, the agent must hypothesize there's some latent parameter/concept governing how people value delayed rewards and then find an appropriate mathematical relationship involving this parameter.
> > > > >
> > > > > > My understanding thus far is that the automated learning process produces models that predict, but are not necessarily similar to the GTs in their functional forms. If it is the symbolic equations being automatically discovered, that certainly changes the whole story.
> > > > >
> > > > > **Yes, agents are producing symbolic equations and mathematical relationships.** See below for concrete evidence from Claude-3.7-sonnet deriving the hyperbolic discounting equation from raw observations.
> > > > >
> > > > > Regarding quantitative evidence of comparing *true model* with the *predicted one*, we deliberately don't compare discovered equations directly to ground truth because: (1) different mathematical formulations can correctly express the same phenomenon (model isomorphism), and (2) no generic metric exists across domains (physics, psychology, ecology).
> > > > >
> > > > > Here is a reasoning trace from claude-3.7-sonnet where it discovers the right equation:
> > > > > ```<thought> I've observed that when offered $10 immediately or $20 in 7 days (a 2x reward for a 7-day wait), the person chose the delayed reward (result = 1).
> > > > >
> > > > > Now I need to predict whether they'll choose $32 immediately or $219 in 208 days.
> > > > >
> > > > > Let me analyze this:
> > > > > - The delayed/immediate ratio is 219/32 ≈ 6.84 (much higher than my observation)
> > > > > - The delay is 208 days (much longer than my observation)
> > > > >
> > > > > In my observation, the person was willing to wait 7 days to double their money. This suggests they have some patience.
> > > > >
> > > > > For the new scenario:
> > > > > - The reward multiplier is much higher (6.84x vs 2x)
> > > > > - But the delay is much longer (208 days vs 7 days)
> > > > >
> > > > > The key question is whether the increased reward ratio compensates for the much longer wait.
> > > > >
> > > > > If we assume a simple hyperbolic discounting model, the person might evaluate the delayed reward as: dR/(1+kD) where k is their personal discount factor.
> > > > >
> > > > > Since they accepted [10, 20, 7], we know that 20/(1+7k) > 10, which means k < 1/7.
> > > > >
> > > > > For the new scenario, they would choose the delayed reward if 219/(1+208k) > 32.
> > > > > Solving: 219 > 32(1+208k)
> > > > > 219 > 32 + 6656k
> > > > > 187 > 6656k
> > > > > k < 0.028
> > > > >
> > > > > Since we estimated k < 1/7 ≈ 0.143 from the first observation, and we need k < 0.028 for them to choose the delayed reward, it's not clear if they will choose the delayed reward.
> > > > >
> > > > > However, since the reward ratio is quite high (6.84x), I think they might still choose the delayed reward despite the long wait. The high reward multiplier might compensate for the long delay.
> > > > >
> > > > > I'll predict they choose the delayed reward.
> > > > > </thought>
> > > > > ```
> > > > >
> > > > > > Further, for the no-prior setting, how would the agent come up with a concept related to cognition?
> > > > >
> > > > > We wanted to clarify the motivation for the no-priors condition. The no-prior condition serves as an important control. Without domain context, agents must discover patterns purely from data; testing whether models can identify underlying regularities without domain-specific biases.
> > > > >
> > > > >
> > > > > > This is by itself a strong hypothesis. Does an established behavioral scientist have the core cognitive capabilities to contribute meaningfully as a mathematician or physicist? I doubt.
> > > > >
> > > > > By "core cognitive capabilities," we mean **necessary but not sufficient** capabilities shared across scientific disciplines. Just as literacy is core to all sciences but insufficient for any specific field, hypothesis formation and experimental design are foundational without being comprehensive. Different fields certainly require additional specialized capabilities; for example, we agree that math and physics may have additional capabilities.
> > > > >
> > > > > > "Designing discriminative experiments that maximize information between theories" This is just statistics, nothing new.
> > > > >
> > > > > While the foundations are indeed grounded in statistics, our results show that **applying these statistical principles to optimally design experiments is far from trivial**, and frontier models fail at this; GPT-4o and Claude-3.7 struggle to design experiments that maximize information gain.
> > > > > In the appendix section, we’ll add more examples of the equations that the models find across domains and clarify in the main text that when we say model discovery, symbolic equation discovery, is a central capability being evaluated.

---

> > > > > > ### Comment · Reviewer_eHZa · 2025-08-06
> > > > > >
> > > > > > "Yes, agents are producing symbolic equations and mathematical relationships."
> > > > > > - The evidence of this, I assume, is this sentence in the sonnet's response: "If we assume a simple hyperbolic discounting model". Can we call this reasoning "the **invention** of an abstract concept" or "the successful **query** of a relevant existing concept"? The word "discover" could loosely mean both, and creates ambiguity. And how about the non-prior case: would sonnet invent the concept of hyperbolic discounting model there as well? After all, these results are not being presented or discussed in the paper.
> > > > > >
> > > > > > "By "core cognitive capabilities," we mean necessary but not sufficient"
> > > > > > - Exactly. Thus the title is too ambitious.
> > > > > >
> > > > > > "applying these statistical principles to optimally design experiments is far from trivial"
> > > > > > - Totally not surprised for an audience who "do science" based on first principles.
> > > > > >
> > > > > > The bottom line is this: The authors argued that an important contribution is a mechanism that "discovers" abstract concepts and symbolic equations. First, this contribution is not being discussed or evidenced in the paper itself. Second, in the one evidence through response, it is not clear if the AI invents abstract concepts (which is a key ingredient of doing science) or fetched an existing concept from its knowledge base.

---

### Official Review · Reviewer_vvLJ · 2025-06-29

**Rating:** 3
**Confidence:** 2

**Summary:**

The paper investigates how LLMs can act like scientists when interacting with an environment. Specifically, it studies whether LLMs can ask questions relevant to a hypothesis to gather new data from the environment, and then use that information to update their understanding about the environment. The process is iterative: the model asks for information about unknowns, receives new data, updates its internal model (posterior), and finally summarises its findings. The paper focuses on two main aspects: (1) how well LLMs can perform as active learners by requesting useful data, and (2) how effectively they update their beliefs and summarise the acquired information.

**Additional Feedback:**

**Some suggestions (related to the Weaknesses)**
- The paper should make the setup and methodology much clearer, possibly with concrete step-by-step examples.
- There should be more discussion of data contamination and how it may affect results.
- Connections to active learning should be explicitly addressed.

**Dataset Code Accessibility:**

Partly

**Dataset Code Comments:**

- The codebase is not well described in the paper or the README, and the paper focuses mostly on the evaluation rather than the dataset itself; currently, the paper doesn't feel like a Benchmark&Dataset paper. Consider explaining more the framework, what can it create, and what is already created in the repository.

**Ethical Considerations:**

No, there are no or only very minor ethics concerns

**Final Justification:**

I thank the authors for clarifying several points. Overall, the paper's novelty is somewhat overstated because the comparison with related literature (e.g., active learning) is not considered or discussed. Further, the paper would benefit from streamlined writing, making it easier to read. As such, incorporating the feedback from all reviewers would improve the paper, and I thank the authors for their efforts. Overall, I keep my score.

**Limitations Weaknesses:**

- The main limitation is clarity: the paper is very hard to follow. The setup is not clearly explained, and trying to understand the full methodology I had to read the appendix, examine the codebase, and carefully analysing the figures. I still don't think I grasped all details. For instance, it's unclear the computation of expected information gain (EIG) are unclear, because it's computed over the context (d), but the context is just a prompt (and not the model's parameters). More clarity and description about all objects in this pipeline is requires, for instance providing an extensive example in the appendix.
- The paper does not discuss active learning, even though it is closely related to it.
- There is not enough discussion or control for data contamination. Since LLMs are trained on large datasets, they may already have seen similar tasks within the training context, which can influence their generalization ability. For instance, it might know the type of distributions used for specific tasks. More discussion here would be useful.
- The codebase README is sometimes out of sync with the actual code, making reproducibility more difficult. Cleaning it, and making it a full example would be helpful.
- To my understanding, there are multiple LLMs used: the Agent which generates questions, but also another LLM that interprets them into a specific query for the environment. Is the same LLM used in both cases?
- The codebase is not well described in the paper or the README; currently, the paper doesn't feel like a Benchmark&Dataset paper. Consider explaining more the framework, what can it create, and what is already created in the repository.

I would consider increasing my score if all these points are carefully addressed.

**Strengths Contributions:**

- The research question is interesting and relevant.
- The paper is an early investigation that helps set a benchmark for what current LLMs can do in this setting.
- The study includes experiments in multiple domains, although these benchmarks mostly represent toy experiments.

---

> ### Author Rebuttal · Authors · 2025-07-31
>
> Thank you for the thoughtful review of our work. We are glad that you found our research direction “interesting and relevant”. We address your comments here:
>
> > I still don't think I grasped all details … it's unclear the computation of expected information gain (EIG) are unclear, because it's computed over the context (d), but the context is just a prompt (and not the model's parameters). More clarity and description about all objects in this pipeline is required… for instance providing an extensive example in the appendix.
>
> We wanted to clarify the computation of EIG. For simplicity, consider a single experiment. Here d represents the experiment that the LLM selects; for example, in item-response theory, d might represent a query about a student’s performance on a particular question on an exam.
>
> In our analyses, we extract the design d from the natural language response of the LLM (the design is formatted with the help of xml tags). Once we parse the design d as a symbolic value, the EIG computation consists of entirely symbolic operations (that do not involve the LLM or natural language. For example, in the example below, we would parse the student id and question id. We then index use the student id and question id to compute the EIG.
>
> \<observe\> [1, 3](student 1 and question 3) \</observe\>
>
> We will clarify this in our revision in Seciton 3.2, and the README (already updated).
>
> > does not discuss active learning
>
> Connection to Active Learning. Active learning enables algorithms to iteratively select the most informative data points for labeling, achieving greater accuracy with fewer labeled instances [Settles, 2009]. This is closely connected to experimental design as an agent needs to design the most informative experiments to reduce infer parameters that the model is most uncertain about. Typically, in active learning, the goal is on collecting useful datapoints for supervised training of a model. In contrast, in OED, the goal is to collect useful experiments in the service of developing a deeper insight about some underlying scientific phenomena.
>
> > There is not enough discussion or control for data contamination.
>
> LLMs likely have seen similar statistical models in training. The "with prior" condition explicitly tests contamination effects where the model can use popular models to simulate the data. Interestingly, prior knowledge often hurts performance, as models fail to revise parameters based on experimentation.
>
> We will discuss contamination explicitly in our discussion section.
>
> > To my understanding, there are multiple LLMs used: the Agent which generates questions, but also another LLM that interprets them into a specific query for the environment. Is the same LLM used in both cases?
>
> We will clarify in Section 4:
> - Scientist LLM: Varies (GPT-4o, Claude, Qwen, etc.) - this is what we evaluate
> - Novice LLM: Always matches the scientist.
> - Environment LLM (only for emotion and morals): Fixed as GPT-4o for consistency across all experiments
>
> > codebase is not well described in the paper or the README … codebase README is sometimes out of sync…a full example would be helpful.
>
> We have revised the README to make the understanding of the codebase and the framework clearer. We will also correspondingly update the paper (expanding section 3.3) to highlight how the abstractions are implemented.
>
> We appreciate your feedback and will incorporate it! Please let us know if you have any other questions.

---

> > ### Comment · Area_Chair_QKgu · 2025-08-05
> >
> > Reviewer vvLJ, please respond to the authors' rebuttal as soon as possible. The author–reviewer discussion period is nearing its end.

---

> > ### Comment · Reviewer_vvLJ · 2025-08-07
> >
> > Thank you for the clarification and discussion. Regarding Active Learning (AL), I continue to believe it is relevant to your work because it aims to select optimal points for a goal -- supervised learning -- whereas this work focuses on a different goal -- "gaining deeper insight into some underlying scientific phenomena". Essentially, both approaches optimise the collection of points with respect to an objective, and I believe that completely omitting any discussion of AL (as it is not mentioned at all in the paper) obscures some of this paper's contributions.
> >
> > Thank you for adding the other clarifications to the paper.

---

### Official Review · Reviewer_5MD4 · 2025-06-30

**Rating:** 3
**Confidence:** 4

**Summary:**

Researchers have been exploring ways of improving LLMs' abilities of doing humans' tasks. One task that researchers themselves have been doing is to make scientific predictions based on experimental data. This benchmark is a gym of testing LLM agents on generating good experimental design and performing scientific prediction. Agents interact with the environment with or without prior knowledge, obtaining data from the environment, and perform inference on the underlying phenomenon. More specifically, the benchmark builds on the framework of Bayesian optimal design and use probabilistic models to construct the environment. LLM agents are expected to perform actions, make scientific predictions, and produce an explanation to teach a novice to make predictions.

The experiments test the prediction error from different LLMs. In general, larger models perform better and closed models outperform open models. There are also some interesting discoveries. The prior knowledge of the problem may not be useful for LLMs to perform well, and the novice model can sometimes outperform the scientist model with only explanations from the scientist model.

**Dataset Code Accessibility:**

Yes

**Dataset Code Comments:**

The complete codes of the benchmark is provided, with instructions of how to interact with the gym.

**Ethical Considerations:**

No, there are no or only very minor ethics concerns

**Final Justification:**

When doing science, a key difference between real-world environments and simulated environments is the availability of the ground-truth models. The benchmark evaluates LLMs in simulated environments using predictive metrics that consider the difficulties of real-world environments, but overlooks the comparison of the underlying models in the simulated environments. This also brings concerns that the unknown underlying model may violate the well-known principle of Occam's razor and still performs well in this benchmark. To compare the models, techniques involving program synthesis may be needed, which is in a different direction.

I see other reviewers also have concerns that the benchmark is not testing the abilities of "doing science". I share the same concern that the scope is too big and the promises are only partly met. Thus I recommend rejection.

**Limitations Weaknesses:**

The biggest weakness of this work is its limited scope. Despite the fact that "scientific theory" is a large topic, the idea of this benchmark focuses on the narrow area of optimal experimental design and probabilistic prediction. I can imagine LLM agents achieve good results by simply generating and executing codes internally for optimal design and probabilistic inference in the future, which seems to be a small improvement contained in the broader scope of solving real world challenges by program synthesis. On the other hand, without the "LLM" components, it may be a good benchmark for experimental design and model discovery. I don't see the reason to couple a benchmark with LLM agents which makes it only useful for LLMs. It would be better to explicitly explain what abilities the benchmark is expecting LLMs to achieve and design metrics to decouple them. For example, beyond a metric of the predictive outcome, I am interested in a few more questions. Does the LLM have a correct model of the scientific problem? Does it understand the uncertainty of the parameters given the observations?

There is a part that defines a Monte Carlo estimator $\mu_\text{NMC}$, but it is only used in Figure 5 to compare different LLMs. I expect that it should also be included in all other figures along with predictive error. The relation between the two metrics could be better explored in this way. On top of predictive error, I also want to see why the models fail. Are they generating an incorrect model? Are they inferencing incorrect parameters? I think these may provide directions to make the pool of metrics richer.

There are noticeable typos which make me concern about the readiness of the work. For example, Figure 2 alone has the following issues:

- The parameters of the functions under the class `Goal` are defined, but parameters in functions for `WorldEnv` and `Agent` are not defined.

- > Will the participant choose \\$10 today of \\$100 in 50 days?

"of" -> "or"

- > so the value of \\$d in D days is 1/1+0.05d.

This seems to be an incorrect formula.

**Strengths Contributions:**

I think the biggest strength of this work is an interactive environment that test LLM agents on a variety of tasks. In real world people do interact with LLMs to gain insights from their data. The work highlights abilities that the current LLMs are missing and advocates for improving on them.

The models in the gym are classic and I am surprised that there have not been an environment that simulates them for downstream evaluation.

---

> ### Author Rebuttal · Authors · 2025-07-31
>
> Thank you for the thoughtful review of our work. We are grateful that you recognize that the benchmark fills an important gap in current research. We address your concerns:
>
> > Limited scope…LLM agents achieve good results by simply generating and executing codes internally for optimal design and probabilistic inference in the future,
>
> We disagree that the task simply reduces to generating and executing code: this framing *overlooks the challenging problem of discovering probabilistic models*. In more detail, generating code for performing optimal experimental design requires *already knowing* the true model structure and parameters: but discovering these is the fundamental challenge!
> This creates a fundamental chicken-and-egg problem:
> - To compute optimal experiments (minimize EIG), you need the generative model $p(y|\theta,d)$
> - To discover the generative model, you need informative experiments
> - In our evaluations, we can compute EIG regret only because we have ground truth - the agent doesn't and part of the challenge is to discover the underlying model.
>
> In real scientific discovery, you cannot write code for optimal design because you don't know what you're trying to discover. Even in our simplified environments, agents must simultaneously: (1) infer model structure from limited data, (2) estimate parameters under uncertainty, (3) select experiments that disambiguate between competing hypotheses, and (4) revise beliefs when data contradicts assumptions. The fact that current LLMs fail at these seemingly simple tasks demonstrates these are non-trivial challenges.
> Further, while our implementation uses probabilistic models to simulate environments, the skills evaluated are fundamental to scientific discovery: model discovery/revision, active experimentation, and scientific communication.
>
> >  small improvement contained in the broader scope of solving real world challenges by program synthesis
>
> We disagree with this assessment. Again, we emphasize that automatically designing experiments and building probabilistic models remains a significant challenge; as evidenced by performance by frontier models. A model that excels at this task would be a valuable tool for accelerating the process of scientific discovery [1].
>
> [1], Lionel, et al. "From word models to world models: Translating from natural language to the probabilistic language of thought." arXiv preprint arXiv:2306.12672 (2023).
>
> > On the other hand, without the "LLM" components, it may be a good benchmark for experimental design and model discovery… I don't see the reason to couple a benchmark with LLM agents which makes it only useful for LLMs
>
> We agree that the benchmark can be easily used for methods beyond LLMs but we focus on LLMs because they represent a promising and timely approach for general scientific reasoning:
> - LLMs have absorbed vast scientific literature, giving them domain knowledge that traditional algorithms lack. Further, LLMs are increasingly competent at coding, and mathematical reasoning, essential for assisting with scientific discovery.
> - LLMs are already being used as assistants by scientists, making evaluation of their capabilities important.
>
> > Does the LLM have a correct model of the scientific problem? Does it understand the uncertainty of the parameters given the observations?
>
> We will add more qualitative examples to the paper, but the main patterns that we notice are: The experiments selected by the agents often have high regret, with respect to the optimal experiment, suggesting that the agent does not select informative experiments. Furthermore, the agent often does not update its initial model; for example, Box’s apprentice uses a restrictive linear model for the population count setting that it does not subsequently update.  Altogether, these results suggest that LMs are strong in hypothesis generation but weak in other components, consistent with the findings in [1,2,3].
>
> [1] Wang, R., Zelikman, E., Poesia, G., Pu, Y., Haber, N., and Goodman, N. D. Hypothesis search: Inductive reasoning with language models. ICLR, 2024.
>
> [2] Qiu et al. Phenomenal Yet Puzzling: Testing Inductive Reasoning Capabilities of Language Models with Hypothesis Refinement. ICLR, 2024.
>
> [3] Piriyakulkij, W. et al. Doing Experiments and Revising Rules with Natural Language and Probabilistic Reasoning. Arxiv preprint.
>
> > (EIG regret)... is only used in Figure 5 to compare different LLMs. I expect that it should also be included in all other figures along with predictive error.
>
> We only included summary statistics in Figure 5 due to constraints on space. We will include complementary plots for prediction error and EIG regret for all figures in the main paper.
>
> > Figure 2 typos
>
> We’ve fixed these errors! Thank you for catching these:
> - "of" → "or"
> - Added parentheses: 1/(1+0.05d)
> - Added parameter definitions for consistency
>
> We appreciate your feedback and will incorporate it! Please let us know if you have any other questions.

---

> > ### Comment · Reviewer_5MD4 · 2025-08-04
> >
> > Thank you for the clarifications. However, I keep my view that the underlying model is important but overlooked in this work, which brings concerns that research in program synthesis will supersede this work. My logic is as follow:
> >
> > (1) For scientific discovery, we care both the prediction and the model parameters. With a correct model, the prediction must be correct. But with a correct prediction, you can't say the model is "correct" scientifically. In the scope of this work, the "model" is the underlying probabilistic program. Without comparing the models, it is hard to say LLMs are doing true science. A further concern is that scientists usually choose the smallest model according to Occam's razor, which is not mentioned in the work at all.
> >
> > (2) But if we compare the models, it means we are testing LLM's ability of generating probabilistic programs, though now in an interactive environment. With the current LLMs, it is possible to let them synthesize programs and their abilities are improving as I type. Also, with a hypothetical probabilistic program, there are no reasons not to let LLMs execute BOED at the the backend because it is mathematically correct to do so. I have no issues with the interactive environment in this work now and I think that is an interesting contribution. But the work may not be on the best route to "doing science automatically".
> >
> > Overall, I think doing "science" is a huge claim not fulfilled in this work.

---

### Official Review · Reviewer_KQMK · 2025-07-03

**Rating:** 4
**Confidence:** 4

**Summary:**

This paper introduces BoxingGym, a benchmark for systematically evaluating large language models (LLMs) on two tightly coupled pillars of scientific discovery: (i) experimental design, quantified by Expected Information Gain (EIG); and (ii) model discovery/explanation, evaluated through a “Scientist → Explainer → Novice” communication protocol. Ten interactive, generative probabilistic environments—spanning psychology, ecology, survival analysis, behavioral economics, etc.—enable principled, quantitative assessment. The authors compare open- and closed-source models (7 B vs 32 B; GPT-4o, Claude, Qwen 2.5, OpenThinker), release all code, and show that larger and instruction-tuned models fare better, yet all models still struggle on both tasks.

**Additional Feedback:**

1. Recommend adding a dedicated Limitations subsection in the main text to foreground synthetic vs real-data gaps and compute costs.
2. Consider incorporating a human evaluation (readability, plausibility) of generated theories to complement Novice error.
3. Move lengthy pseudocode/math derivations to the appendix to streamline the main narrative.
4. Future work could integrate real laboratory or crowd-sourced behavioral datasets to test external generalization.

**Dataset Code Accessibility:**

Yes

**Dataset Code Comments:**

The repository includes an environment.yml and one-click scripts; adding average GPU-hours per experiment run in the README would further help researchers gauge resource requirements.

**Ethical Comments:**

The submission operates exclusively on synthetic, probabilistically-simulated data and releases only open-source code and environment definitions; no personally-identifiable information, human subjects, or sensitive real-world data are involved.

**Ethical Considerations:**

No, there are no or only very minor ethics concerns

**Limitations Weaknesses:**

1. Prior-knowledge effect under-analyzed: claims that “no prior performs better” lack quantitative ablation across prior strengths.
2. Compute cost opaque: Nested Monte Carlo EIG is expensive; GPU-hours and memory footprints are not reported.
3. Single-dimension explanation metric: Novice error captures utility but ignores readability or conciseness of generated theories.
4. Figure clarity: several plots do not specify whether error bars are ± 1 σ or 95 % CI; legends for line styles are hard to read.
5. Synthetic-only validation: all environments are simulated; no demonstration on real experimental datasets, limiting external validity.

**Strengths Contributions:**

1. First holistic benchmark that evaluates both experimental design and model discovery in a single framework—filling a clear gap.
2. Broad scientific coverage: ten well-motivated domains with generative simulators ensure diversity and realism.
3. Sound information-theoretic metric (EIG) for experiment quality, grounded in Bayesian optimal experimental design.
4. Comprehensive empirical study: ablates model size, prior knowledge, instruction- vs reasoning-finetuning.
5. Fully open-sourced and reproducible: code, data, and scripts meet NeurIPS reproducibility standards.

---

> ### Author Rebuttal · Authors · 2025-07-31
>
> Thank you for your thoughtful review and feedback on our work. We are thankful for your recognition that this is the “first holistic benchmark” for experimental design and model discovery. We address your main concerns here.
>
> > Compute costs
>
> We've added detailed resource requirements:
> - Agent Experiments Per experiment: All experiments run on a single A40 (40GB) with VLLM. For 3B models this takes around
> - Full benchmark (with/without prior + communication) run with 3B model: ~5 GPU hours
> - EIG computation: While EIG computation is expensive, it is extremely parallelizable across cores and can be run asynchronously with running the benchmark with an agent.
> ~10 secs per design with 1000 samples used for the estimate (parallelizable)
>
> > Prior-knowledge effect under-analyzed: claims that “no prior performs better” lack quantitative ablation across prior strengths.
>
> While this is an interesting question, it is difficult to vary the prior strengths across different experiments.
> In future work, we could explore more explicit ways of endowing neural networks with specific priors through finetuning on data generated from prior distributions [1].
>
> [1] [2305.14701] Modeling rapid language learning by distilling Bayesian priors into artificial neural networks
>
> > Single-dimension explanation metric: Novice error captures utility but ignores readability or conciseness of generated theories.
>
> We acknowledge this limitation and have added a couple of subjective metrics, like readability by using a stronger llm (gpt-4.1) as a judge. For conciseness, we use the proxy of word count. For example, here ratings for readability and conciseness. Here are the results for the death process environment.
>
> Model | Conciseness (Word Count) | Readability (1-5)
> --------| --------| --------|
> gpt-4o|  190.9 words |  4.00 |
> claude-3.7-sonnet | 250.0 words |  4.00 |
> Openthinker-7b | 368.5 words | 3.20 |
> Qwen2.5-7b instruct | 171.8 words | 3.90 |
>
> We will add the complete results and human agreements with llm scores on a small subset in the final version.
>
> > Figure clarity
>
> All figures use the standard error. We will add it to Figure 5, 6 where this is missing.
>
> > Synthetic-only validation
>
> BoxingGym is the first step towards more holistic evaluation of LLM agents with experimental design, discovery and communication section acknowledging this. We chose synthetic environments to start with because:
> - It enables controlled evaluation with ground truth
> - Allows for principled EIG computation
> - Real experimental data is expensive and would require running real experiments — for example, running experiments with humans on prolific with thousands of humans with hundreds of different stimuli.
>
> And crucially we’d have to run new experiments for each unique set of experimental choices from a user of our benchmark.
> We will acknowledge this limitation more clearly in the revision. Future work will incorporate real-world experimental data.
>
> Please let us know if you have any other questions.

---

> > ### Comment · Area_Chair_QKgu · 2025-08-05
> >
> > Reviewer KQMK, please respond to the authors' rebuttal as soon as possible. The author–reviewer discussion period is nearing its end.

---

### Official Review · Reviewer_jjig · 2025-07-03

**Rating:** 5
**Confidence:** 4

**Summary:**

This paper introduces a benchmark for scientific discovery with LLMs, where the agent conducts experiments in an environment defined by a probabilistic generative model in order to reduce uncertainty about the parameters of the model. It then generates a text explanation for a novice agent which is required to make predictions about the system.

**Dataset Code Accessibility:**

Yes

**Ethical Considerations:**

No, there are no or only very minor ethics concerns

**Final Justification:**

The other reviewers bring up salient weaknesses, but I still think that the core ideas behind the benchmark, eg distillation of scientific knowledge via communication, are interesting and valuable.

**Limitations Weaknesses:**

Various questions/suggestions:
(1) It would be very useful to have an appendix with detailed examples of how generated text interfaces with the probabilistic model. It's not clear right now how the design space for experiments is instantiated or described to the model, or how natural language inputs are then converted into experiments.

(2) At what rate do the novice agents correctly implement the explanations from the scientist? E.g., in the example in the paper when the hint is "The value is hyperbolically
discounted with a factor of 0.05, so the value of $d in D days is 1/1+0.05d", can you check whether the novice agent makes predictions according to this model? Measuring improvement in performance from explanations is a nice idea, but the connection would be tighter if it's possible to quantify the extent to which explanations are used, or whether they're incorrectly interpreted by the novice, overriden by parametric knowledge, etc.

(3) It would be informative to have a baseline for the novice's performance without the hint. This depends on what the novice is told about the domain, which would be helpful to clarify in the paper. If the novice is given a similar general description of the domain as the scientist (in the setting where this prior information is provided), then zero-shot predictions from the novice without explanation would be helpful context for the amount of improvement.

**Strengths Contributions:**

The paper introduces an exciting and ambitious direction. Systematically testing the experimentation and theory-building capabilities of LLMs is clearly very important. Integrating experimentation with communication as a test for the development of a concise theory is a compelling way to develop a tractable sandbox for the scientific process, and a creative idea. The fact that LLMs display uneven performance even in domains where the generative model follows a relatively simple parametric form is quite interesting and suggests that there is room for this kind of benchmark to guide model improvements. I support acceptance.

---

> ### Author Rebuttal · Authors · 2025-07-31
>
> Thank you for your supportive review of our work! We're glad you found our paper “exciting and ambitious”. We address your questions:
> > Detailed examples of how the text interfaces with the probabilistic model
>
> We'll added comprehensive examples in Appendix E showing the complete pipeline. For instance, in the IRT environment:
> - **Probabilistic Model**: Samples from Bernoulli(logit(α₂ - β₃)) where α₂ is student ability and β₃ is question difficulty
> - **NL Input**: "Observe if student 2 answers question 3 correctly <observe>2, 3</observe>"
> - **Conversion**: The environment parses the xml to extract (student_id=2, question_id=3)
> - **Environment**: The environment then responds if the student answered the question correctly (“1”) or incorrectly (“0”).
>
> > Do novice agents correctly implement explanations?
>
> Measuring if the novice agents faithfully implement explanations is difficult. To get an idea of how accurately agents are able to implement explanations, we ask an LLM to judge the faithfulness of the reasoning to the provided explanation. We prompt `gpt-4.1` as a judge, asking it to rate faithfulness from 1-5. We find the following results for gpt-4o as a scientist and novice:
>
> | Environment                    | Model              | Faithfulness Score |
> |--------------------------------|--------------------|--------------------|
> | **DEATH_PROCESS**              | gpt-4o             | 4.80               |
> |                                | OpenThinker-7B     | 2.10               |
> |                                | qwen2.5-7b-instruct| 2.00               |
> | **DUGONGS**                    | gpt-4o             | 4.00               |
> |                                | OpenThinker-7B     | 2.50               |
> |                                | qwen2.5-7b-instruct| 1.60               |
> | **EMOTION**                    | gpt-4o             | 4.20               |
> |                                | OpenThinker-7B     | 2.80               |
> |                                | qwen2.5-7b-instruct| 2.40               |
> | **HYPERBOLIC_TEMPORAL_DISCOUNT**| claude-3-7-sonnet-20250219 | 3.90     |
> |                                | gpt-4o             | 3.50               |
> |                                | OpenThinker-7B     | 1.60               |
> |                                | qwen2.5-7b-instruct| 2.80               |
> | **MORALS**                     | gpt-4o             | 4.00               |
> |                                | OpenThinker-7B     | 1.60               |
> |                                | qwen2.5-7b-instruct| 2.40               |
>
> In general, models that have been instruction tuned faithfully follow the explanations. Smaller models that have been overoptimized for math and code struggle with these, getting lower faithfulness scores. We will add complete results to the appendix in the revision.
>
> > If the novice is given a similar general description of the domain as the scientist (in the setting where this prior information is provided), then zero-shot predictions from the novice without explanation would be helpful context for the amount of improvement.
>
> Thanks for your question. To clarify, what you describe (novice given similar general description) equivalent to the zero-shot performance of the scientist (before any experimentation); the novice receives the same information about the environment as the scientist in the *with priors* condition. So, the 0-shot performance of the scientist (before any experimentation) are the same as the performance of the novice without the hint / explanation from the scientist. We provide the comparison below (for claude-3.7-sonne for a few environments) and will add plots to section 4.2 to make the point clearer.
> |Environment | Novice with no explanation | Novice with explanation | Difference
> |----------|-----------| --------- | ---------
> | hyperbolic_temporal_discount  | 0.3181         | 0.7869           | -0.4688    |
> | location_finding              | 0.2981         | 13.9702          | -13.6721   |
> | death_process                 | 0.5359         | -1.0829          | 1.6188     |
> | irt                          | 0.1200         | 0.1200           | 0.0000     |
> | dugongs                      | -0.0405        | -0.0579          | 0.0174     |
> | survival                     | 0.0439         | 1.0009           | -0.9570    |
> | peregrines                   | 1.9523         | -0.6524          | 2.6047     |
> | morals                       | 0.4010         | 0.6843           | -0.2833    |
> | emotion                      | 1.0397         | 0.9049           | 0.1348     |
> | lotka_volterra               | 0.3768         | -0.0078          | 0.3846  |
> We see that for some environments the explanations from the scientist seem to hurt the model over just using a general description of the environment.
>
> We will update our paper to more clearly articulate these points. Please let us know if you have any other questions.

---

### Decision · Program_Chairs · 2025-09-18

**Decision:**

Reject

**Comment:**

This is a highly controversial submission. The benchmark includes 10 scientific problems operationalized as probabilistic programs, and a quantitative paradigm for testing LLMs as scientists conducting experiments, making predictions, and building verbal explanations of these environments. The experiments are evaluated by expected information gain, and the explanations are evaluated by the prediction error of student models presented with them. Importantly, the scientist/teacher models are not presented with the underlying unknown variables and probabilistic models; hence this work departs from traditional optimal experimental design.

Two reviewers supported acceptance (with scores of 4 and 5) and three opposed it (scores of 2, 3, and 3). I have read the paper, the reviews, and the authors’ rebuttal. The criticisms of the two most critical reviewers, eHZa and 5MD4, reflect a different view of what “doing science” amounts to. The paper takes an instrumentalist approach, where a scientific explanation is deemed good if it induces correct predictions. By contrast, these reviews adopt a scientific-realist approach, where a good explanation is one that correctly captures underlying relations and entities (5MD4: “Does the LLM have a correct model of the scientific problem?”; eHZa: “derivation of governing equations from first principles and data” as a desideratum). Both the instrumentalist goal (prediction) and the realist goal (model selection) are legitimate; this submission explicitly targets the former, and I evaluate it on that basis. Benchmarks pursuing the realist goal would also be valuable, but they are outside the scope of the submission.

Creating a benchmark for emulating scientific discovery with LLMs is undoubtedly an important direction. This contribution is, in my view, sufficiently complete to make a substantial and highly original first step in this direction, justifying a NeurIPS D&B paper. I therefore recommend acceptance as a poster, aligning with the minority view. Reviewer vvLJ’s comments on clarity and context are important to address in the camera-ready version.

===== FINAL UPDATE FROM DB Track PCs ====

The final decision for this paper has been taken by the program chairs after consultation with the SACs. All Senior Area Chairs have ranked papers according to the feedback from the AC during the review process. We decided to leave the original meta-review to reflect the opinion of the AC in light of the initial discussions with reviewers and SAC.